# OpenReview forum: "PoisonedEye: Knowledge Poisoning Attack on Retrieval-Augmented Generation based Large Vision-Language Models"
_ICML.cc/2025/Conference — ICML 2025 poster_

### Official Review · Reviewer_vyyS · 2025-03-11

**Overall Recommendation:** 4

**Summary:**

This paper presents a knowledge poisoning attack targeting MuRAG systems used for vision language models.
It introduces a method to manipulate MuRAG system outputs by injecting poisoned image-text pair into the multimodal knowledge base.
This work extends the textual RAG attack to multimodal setting showing the vulnerability of VLM relying on external multimodal knowledge base.
It presents three attack strategies: baseline, single query targeted attack, and class query targeted attack, which extends the attack to an entire class of queries by optimizing for class-based retrieval.

**Claims And Evidence:**

- this is the first work to study multimodal poisoning attacks on MuRAG system
- Experiments are supported by Comprehensive evaluation on multiple datasets using differetn retrievers and LVLMs
- Ablation studies are thoroughly exaimed factors affecting attack effectiveness
- Strong empirical results

**Essential References Not Discussed:**

NA

**Experimental Designs Or Analyses:**

Yes, the experiment setting, baseline, retriever, datasets.

**Methods And Evaluation Criteria:**

Yes, the dataset and knowledge base (OVEN) makes sense.
The retriever part can be improved such as using multimodal retriever (e.g., UniIR). it seems currently it is image-only retriever.

**Other Comments Or Suggestions:**

Related work on attack to VLM with poisoned image: Can Language Models be Instructed to Protect Personal Information? https://arxiv.org/abs/2310.02224

**Other Strengths And Weaknesses:**

Strengths: minimal poisoning requirement, this method requires only a few sample to successfully alter the VLM response, making it highly efficient.
Weaknesses: assume a controllable knowledge database which might not be practicel in well-secured system with strict data integrity mechanisms; defensive strategies not explored

**Questions For Authors:**

1. How to defend such attack? The current work focus on exploring this attack
2. How would attacker attack a private multimodal database? It seems the current attack requires access to insert data into a database, how does such attack work in real world setting where people/service host a private database or just use the web.

====post rebuttal =====
Thanks for answering questions. Clear now

**Relation To Broader Scientific Literature:**

The key contribution is studying the poisoning attack to MuRAG system. Recent work focus on text only RAG system.

**Theoretical Claims:**

NO

---

> ### Author Rebuttal · Authors · 2025-04-01
>
> We sincerely thank your constructive comments.
>
> > Q1. The retriever part can be improved such as using multimodal retriever (e.g., UniIR).
>
> We conduct additional experiments specifically on the UniIR_CLIP_SF retriever model, which is  the best-performing model in the UniIR framework. As the results shown in the table below, our attack achieves 99.45% PSR on the UniIR model, demonstrating our method's effectiveness on different retriever models.
>
> |   Retriever   | RSR-1  | RSR-K  |  ARD   |  PSR   |
> | :-----------: | :----: | :----: | :----: | :----: |
> | UniIR_CLIP_SF | 98.90% | 99.72% | 0.7556 | 99.45% |
>
> \* This experiment is conducted on LLaVA-v1.6-Mistral 7B LVLM, and Places-365 dataset.
>
> > Q2. How to defend such attack? The current work focus on exploring this attack.
>
> We conduct additional experiments on the following possible defense strategies to demonsrate the effectiveness of our attacks. 1) add noise on poison image; 2) random crop poison image; 3) apply RoCLIP [1] that rematches every image with the text that is most similar to it in the database for retrieved samples. As the results shown in the table below, noise and random crop reduce the attack PSR to some extent. However, even after the random crop defense, the PSR remains at 45.20%, indicating that nearly half of the attacks are still successful. For RoCLIP, the defense is effective for the original attack. However, this defense can be easily bypassed by an enhanced attack that maximizes the poison image-text relation in the poison crafting process. As shown in the last line of the table, the RoCLIP only reduce the enhanced attack PSR by 38.11%, indicating that our poisoning attack framework can not be effectively defended so far.
>
> |           Defenses            | RSR-1  | RSR-K  |  ARD   |  PSR   |  ∆PSR   |
> | :---------------------------: | :----: | :----: | :----: | :----: | :-----: |
> |          No Defense           | 85.09% | 92.95% | 0.7603 | 92.63% |    -    |
> |       Noise (max = 16)        | 54.52% | 64.93% | 0.7612 | 64.65% | -27.98% |
> |   Random Crop (scale = 0.7)   | 32.32% | 45.20% | 0.7597 | 45.20% | -47.43% |
> |            RoCLIP             | 83.28% | 91.78% | 0.7609 | 0.00%  | -92.63% |
> | RoCLIP (with enhanced attack) | 54.24% | 71.78% | 0.8325 | 54.52% | -38.11% |
>
> \* This experiment is conducted on Siglip-so400m retriever,  LLaVA-v1.6-Mistral 7B LVLM, and Places-365 dataset.
>
> > Q3. How would attacker attack a private multimodal database? It seems the current attack requires access to insert data into a database, how does such attack work in real world setting where people/service host a private database or just use the web.
>
> In most existing literature on poisoning attacks, the following real-world settings are commonly considered, where poisoning attacks can work for a private database: 1) Direct database access (e.g., insiders or hackers); 2) Malicious data vendors. Injecting poison samples into datasets sold to victims; 3) Public data poisoning. Attackers publish poison data on the Internet, which victims unknowingly collect to build their private databases. Therefore, our paper adopts a consistent setting for the attacker, aligning with existing poisoning attacks.
>
>
>
> Referneces
>
> [1] Robust contrastive language-image pretraining against data poisoning and backdoor attacks. NeurIPS'23.

---

### Official Review · Reviewer_ZyNQ · 2025-03-12

**Overall Recommendation:** 3

**Summary:**

This paper proposed a poisoning attack against Multi-Modal RAG systems, especially for LVLM RAG systems. The paper formulates the goal as an optimization problem and discusses to solve it in two different settings: Single Query Targeted Attack and Class Query Targeted Attack. Given a target image-text pair, the proposed method could inject only a single image-text pair into the knowledge database to induce the RAG system to output a target response defined by the attacker. The paper conducts a comprehensive evaluation and the proposed attack is effective in achieving the attack goal.

**Claims And Evidence:**

The claims are supported.

**Essential References Not Discussed:**

There are no missing references.

**Experimental Designs Or Analyses:**

Yes I checked the experiment settings, evaluation metric design, evaluation settings, and datasets.

 Due to space limits, the ablation study does not include many aspects. For instance, the ablation study on $\alpha$, $s$, and $\epsilon$ is missing.

**Methods And Evaluation Criteria:**

make sense.

**Other Comments Or Suggestions:**

The ablation study could contain more aspects, e.g. test more retrievers and LLMs, and those hyperparameters of the proposed method. There may not be enough space for that but These results could be included in the Appendix.

**Other Strengths And Weaknesses:**

* Strengths:
	* Good writing to express the core ideas. The illustration figure is also clear to understand.
	* Adapting textual RAG poiong attack to MultiModal scenario is a practical and important direction to explore. This paper applied RAG attack to LVLM RAG systems effectively and provides potential research directions for future exploration.
	* Experiment design is very good and clear, especially the Evaluation Metrics of RSR-1, RSR-K, ARD and PSR.
* Weaknesses:
	* The attacker's capability is too strong that the attacker could attack specific query and image. Alghough the Class Query Targeted Attack is proposed to discuss the image part, the text part should also be included to make the attack more practical and expand the attack scope. In Appendix B the paper discusses attacking several queries, but this scenario is not as general as the Class Query Targeted Attack for the image.
	* The paper only discusses LVLMs and image-text tasks. So the writing should not focus on stressing "the first poining attack designed for MuRAG", where MuRAG could include other modalities (e.g., audio) and a broader scope of tasks.
	* The paper does not explore the impact of the similarity metrics used in the LVLM RAG systems. Or the reason for using L2-distance is not justified.
	* Based on the experimental results, when the Retrieval Number K increases, there is a significant drop in PSR. These results indicate that the proposed attack could be easily defended by increasing the retrieval number.  Defenses can be explored.

**Questions For Authors:**

1. Could the authors further explain the drop in PSR when retrieval number K increases? Does this mean the proposed attack could easily be defended by increasing the retrieval number? And could this issue be solved?

2. Since there is a well-defined Class Targeted Attack setting for the image part, why cannot also include a similar attack setting for the text part (rather than just some samples in the Appendix)? Is there any challenge?

**Relation To Broader Scientific Literature:**

This paper is an adaptation of RAG poisoning attacks in the LVLM domain. It applied attack techniques from textual RAG (Zou et al., 2024; Chen et al., 2024b; Cheng et al., 2024) to MuRAG, especially focusing on text-image tasks. The proposed method is effective in the text-image domain and inspires new research directions.

Zou, W., Geng, R., Wang, B., and Jia, J. Poisonedrag: Knowledge corruption attacks to the retrieval-augmented generation of large language models. *arXiv preprint* *arXiv:2402.07867*, 2024.

Chen, Z., Xiang, Z., Xiao, C., Song, D., and Li, B. Agentpoison: Red-teaming llm agents via poisoning memory or knowledge bases. *arXiv preprint arXiv:2407.12784*,2024b.

Cheng, P., Ding, Y., Ju, T., Wu, Z., Du, W., Yi, P., Zhang, Z., and Liu, G. Trojanrag: Retrieval-augmented generation can be backdoor driver in large language models. *arXiv* *preprint arXiv:2405.13401*, 2024.

**Theoretical Claims:**

NA

---

> ### Author Rebuttal · Authors · 2025-04-01
>
> We sincerely thank your constructive comments.
>
> > Q1. The ablation study on α, s, and ϵ is missing.
>
> We conduct additional ablation studies on α, s, and ϵ as the table shown below.
>
> |   α   | RSR-1  | RSR-K  |  ARD   |  PSR   |
> | :---: | :----: | :----: | :----: | :----: |
> |  0.1  | 43.01% | 54.79% | 0.8463 | 54.52% |
> | 0.01  | 83.83% | 92.87% | 0.7619 | 92.32% |
> | 0.001 | 80.54% | 89.04% | 0.7700 | 88.76% |
>
> |  s   | RSR-1  | RSR-K  |  ARD   |  PSR   |
> | :--: | :----: | :----: | :----: | :----: |
> |  10  | 56.43% | 69.31% | 0.8207 | 69.04% |
> |  20  | 67.12% | 79.17% | 0.8033 | 78.90% |
> |  50  | 78.08% | 89.31% | 0.7769 | 89.04% |
> | 100  | 84.10% | 92.60% | 0.7609 | 92.05% |
>
> |  ϵ   | RSR-1  | RSR-K  |  ARD   |  PSR   |
> | :--: | :----: | :----: | :----: | :----: |
> |  4   | 48.76% | 61.64% | 0.8341 | 61.36% |
> |  8   | 68.76% | 78.90% | 0.8001 | 78.63% |
> |  16  | 82.46% | 92.60% | 0.7621 | 92.05% |
> |  32  | 92.32% | 97.26% | 0.7268 | 96.71% |
>
> \* All experiments are conducted on Siglip-so400m,  LLaVA-v1.6-Mistral 7B, and Places-365.
>
> > Q2. Why cannot also include a similar Class Query Targeted attack setting for the text part? Is there any challenge?
>
> Under the class query targeted attack setting for texts, the attack should be activated when user asks similar questions to the target text. However, the inducibility of the poison text becomes very weak when user asks different questions, as shown in the rebuttal of reviewer adRg Q4. This is mainly due to the black box assumption of the VLM, which makes it hard to alter the response by crafting poison prompts. Therefore, enhancing attack capability on text side under the class query targeted attack scenario will be a valuable research direction for future works.
>
> > Q3. The writing should not focus on MuRAG, which could include other modalities (e.g., audio).
>
> Thanks for your comment. We will revise the use of "MuRAG" in the final version of our paper.
>
> > Q4. The reason for using L2-distance as similarity metric is not justified.
>
> Current works on RAG like DPR [1], UniIR [2], FAISS [3] mainly employ inner product and L2-distance as similarity search metrics. The L2-distance has equivalent effect as the inner product, because it can be mathematically transformed to the L2-distance by the equation shown below when the embeddings are normalized. A small L2-distance indicates closer proximity and stronger feature similarity. Therefore, the L2-distance metric is reasonable for retrieval process.
> $$
> L_2(v_1,v_2) = ||v_1-v_2||_2 = \sqrt{||v_1||^2+||v_2||^2-2v_1⋅v_2} = \sqrt{2-2InnerProduct(v_1,v_2)}
> $$
>
> > Q5. Could the authors further explain the drop in PSR when retrieval number K increases? Does this mean the attack could be defended by increasing retrieval number? Could this issue be solved?
>
> As retrieval number K increases, more clean samples from the database is retrieved and integrated into the prompt, then the VLM is provided with more information that could lead to the correct answer. Therefore, the VLM is less likely to produce the target answer and the PSR decreases.
>
> There are indeed some drop in PSR as retrieval number K increases. However, this issue can be solved by increasing poison injecting numbers. In our main experiments, we only inject one poison sample into the knowledge database. We conduct additional experiments with more poison samples when K=8. As the results shown in the table below, the PSR increases when poison number is larger than 1, even when poison number is 2, the PSR holds at 85.20%, solving the PSR dropping issue.
>
> | Poison Number | RSR-1  | RSR-K  |  ARD   |  PSR   |
> | :-----------: | :----: | :----: | :----: | :----: |
> |       1       | 83.56% | 95.06% | 0.7613 | 51.78% |
> |       2       | 83.83% | 95.61% | 0.7615 | 85.20% |
> |       4       | 84.65% | 95.89% | 0.7614 | 81.09% |
>
>
>
> References
>
> [1] Dense Passage Retrieval, EMNLP'20
>
> [2] UniIR, ECCV'24
>
> [3] The Faiss library, arXiv:2401.08281

---

### Official Review · Reviewer_adRg · 2025-03-13

**Overall Recommendation:** 3

**Summary:**

The paper proposes the first knowledge poisoning attack against MuRAG system. The core contribution includes three attack variants (PoisonedEye-B, PoisonedEye-S, PoisonedEye-C) that span single-query and class-query targeted attack.

**Claims And Evidence:**

The claims made in the submission are generally supported by clear and convincing experimental evidence. There are a few concerns:

1. I would like to see more attack baselines in RAG and how well they work in MuRAG compared to MuRAG.
2. For baseline attack and PoisonedEye-S, I think the author made an assumption that the attacker knows the target image in the query, but was not made explicit. I would like to see the experimental results of the attack behaving when the poisoned image is not exactly the same as the target image.

**Essential References Not Discussed:**

Key related works are all discussed.

**Experimental Designs Or Analyses:**

The experimental designs are mostly sound.

**Methods And Evaluation Criteria:**

Both the methods and the evaluation criteria make sense.

**Other Comments Or Suggestions:**

N/A

**Other Strengths And Weaknesses:**

Strengths:

1. The writeup is very clear, with good motivation.
2. Novelty and significance as the first poisoning attack explicitly targeting MuRAG systems.
3. Comprehensive experimental validation across multiple LVLMs and retrievers.

Weakness:

1. Limited exploration of practical real-world scenarios for the baseline and single-query targeted attacks.

2. Lack of extensive discussions or experiments addressing potential adaptive defenses.

**Questions For Authors:**

1. In the class query attack, how is the "class" defined, especially considering that CLIP is trained on caption datasets? How does the class query attack perform on datasets like COCO, where classes are based on captions rather than predefined categories?

2. How are the images selected for the class query attack? Are they chosen from the training dataset, or are they randomly selected from the internet? What is the process for ensuring these images are representative of the class?

3. While the class query attack assume differently, the baseline and single query attacks, which are two separate attacks, assume that the poisoned image is identical to the user's query image. How valid are these results in real-world scenarios where the user's image might be from the same class but not identical? Could you provide experiments using images from the same class to evaluate the retrieval effectiveness in such cases?

4. The current attacks optimize only the image component. How does the text influence the attack, especially if the text is very different from the query? Could you extend the optimization to include text and conduct experiments where the text distance becomes a significant factor (e.g., user asking very different questions with similar images)?

5. Are there any adaptive defense strategies that could mitigate the class query attack, especially against minor image modifications or augmentations like random noise? For example, could techniques like RoCLIP [1], which swaps image representations with nearest neighbors, be effective in defending against such attacks?

6. Are there any other baselines or related work that could be compared to, such as the textRAG attack mentioned in the related work? How does PoisonedEye compare to these existing methods in terms of effectiveness and robustness?

[1] Yang, Wenhan, Jingdong Gao, and Baharan Mirzasoleiman. "Robust contrastive language-image pretraining against data poisoning and backdoor attacks." Advances in Neural Information Processing Systems 36 (2023): 10678-10691.

**Relation To Broader Scientific Literature:**

The paper builds on top of the existing RAG attack on text and extend the idea into multimodal contexts.

**Theoretical Claims:**

There are no theories presented in the paper.

---

> ### Author Rebuttal · Authors · 2025-04-01
>
> We sincerely thank your constructive comments.
>
> > Q1:  "Class" definition & class query attack perform on captions datasets like COCO
>
> The "class" in this context denotes a group of images that have similar semantic meanings (i.e., close L2 distance on pre-trained encoders like CLIP). For image classification datasets, we assume that they pre-classify semantically similar images into groups and attack images in the same class of the target image. For image caption datasets like COCO, our attack will be also effective on semantically similar images of the target image.
>
> For example, we randomly select a target image and evaluate our attack for its semantically similar images (e.g., with 100 closest images on CLIP image distance) in COCO dataset. As the result shown below, the PSR holds at 67.39%, demonstrating our attack effectiveness on semantically similar query images in caption datasets like COCO.
>
> | Dataset | RSR-1  | RSR-K  |  ARD   |  PSR   |
> | :-----: | :----: | :----: | :----: | :----: |
> |  COCO   | 93.42% | 98.35% | 0.7298 | 67.39% |
>
> \* This experiment is conducted on Siglip-so400m, and  LLaVA-v1.6-Mistral 7B.
>
> > Q2: How are the images selected for the class query attack? How to ensure that these images are representative of the class?
>
> We aims to alter the system's response for query images that have similar semantic meanings to the target image. Therefore, regarding the target image as the center, we find its semantic neighbors (images with close L2 distance) from an auxiliary dataset (WebQA) by measuring the CLIP distance and select them as representative samples of the "class" centered by the target image.
>
> > Q3: How valid are the results of baseline and single query attack in real-world scenarios where the user's image is not identical?
>
> In our main evaluation, we adopt class query attack assumption where the target image is not identical to the user's query image. The results in Table 1 and Figure 2 of our paper is exactly what you are looking for. For the baseline and single query attack assumption where the target image is identical to the user's query image, the experiments is conducted in Appendix.D.
>
> > Q4:  Text influence on the attack &  experiments about text distance
>
> For the text part, we conduct additional experiments to evaluate the attack effectiveness when user asks similar questions with similar images to the target. For initial target text = "Which scene category does this image belong to?", we add 4 relevant target texts and test three other user queries that are highly relevant, medium relevant, and low relevant (marked by an LLM) to the target text as the table shown below. The results show that our attack can succeed in certain cases when the user query is relevant to the target query. When the user asks irrelevant questions, the attack does not activate as expected with PSR=2.19%. Therefore, enhancing attack capability on text side under the class query attack scenario will be a valuable research direction for future works.
>
> |                    Query                     | Distance to target text | Relevance to target text | RSR-1  | RSR-K  |  ARD   |  PSR   |
> | :------------------------------------------: | :---------------------: | :----------------------: | :----: | :----: | :----: | :----: |
> |    Which scene does this image represent?    |         0.7414          |           High           | 56.16% | 69.04% | 0.8019 | 38.08% |
> | Can you identify the environment shown here? |         0.8251          |          Medium          | 31.23% | 43.83% | 0.8023 | 20.00% |
> |        What is shown in the picture?         |         0.8739          |           Low            | 41.09% | 56.16% | 0.8013 | 2.19%  |
>
> > Q5:  Are there any adaptive defense strategies (e.g., random noise and RoCLIP)?
>
> We conduct experiments on defense strategies including random noise, random crop, and RoCLIP to demonsrate the effectiveness of our attack. The experimental results show that our poisoning attack framework can not be effectively defended so far. Please refer to the rebuttal of Reviewer vyyS, Q2 for details.
>
> > Q6: Are there any other baselines or related work that could be compared to?
>
> There are indeed some studies [1-3] on text RAG attacks. For PoisonedRAG [1], we have adapted it to the vison-language modality as our baseline method PoisonedEye-B. The comparison between our baseline and our proposed methods has been shown in Table 1 and Figure 2 of our paper. The results show that our proposed methods outperform the baseline in all cases. TrojanRAG [2] and AgentPoison [3] are remotely related with our work but focus on different attack settings and tasks. e.g., they both requires the attackers have the capability to modify user queries and [3] studies the LLM agent task. This is not consistent with our poisoning attack threat model, where attackers cannot modify user queries.
>
>
>
> [1] PoisonedRAG, USENIX'25
>
> [2] TrojanRAG, arXiv:2405.13401
>
> [3] AgentPoison, NeurIPS'24

---

> > ### Comment · Reviewer_adRg · 2025-04-03
> >
> > Hi, thank you so much for your responses! I have a few follow-up questions on your answers:
> >
> > 1. For Q2, did you use any distance metrics between the target image and the query image, or did you select the target image randomly from the dataset? The difference being that, the former case assumed a known query image, while the latter doesn't. My concern is that, the user may query some images that, while sharing the semantic meaning, are less representative in the target class. In that case, I am wondering how effective would the method be.
> >
> > 2. For Q3, my point was that this might be less of a contribution, as the assumptions of the query and target image being the same is too strong. The class query attack has a reasonable assumption, but I don't think the single-image attack's assumption is realistic.
> >
> > 3. For Q4, (1) is this class type attack or single image attack? (2) I am wondering why the PSR is very low, even with texts that are very relevant to the initial questions? (compared to the result in paper which generally has a higher ASR of 60%~70%.
> >
> > 4. For RoCLIP, I am wondering for the enhanced attack, did you conduct it on single-image or class attack? Specifically, I am wondering with additional processing, if the image could still be representative of the class if it is a class-type attack? If it a single-image attack, like I mentioned before, I think the results are not as valid as the attack assumptions are too strong.

---

> > > ### Author Response · Authors · 2025-04-08
> > >
> > > We sincerely thank you for your constructive comments and thoughtful feedback. Regarding the follow-up comments, we provide the following responses.
> > >
> > >
> > >
> > > A1. Thank you for your thoughtful suggestion. Our main experiments utilize the latter case you mentioned, where both target and query images were randomly selected from the same class in a classification dataset, without considering their representativeness. Therefore, the current results include query images that are both representative and non-representative of the class, and the final PSR is an overall average of all possible query images from the same class.
> > >
> > > For the former case you mentioned where the user may query some images that are less representative in the target class, we also conduct additional experiments to evaluate such scenarios. In detail, for each class, we select the least representative image (i.e., the one with the largest distance to the class center) to evaluate the effectiveness of our class attack. According to the results shown in the table below, the PSR holds at 81.09% for these images, demonstrating that our attack reamins effective for query images that share semantic meaning with the class, even if they are less representative. We will incorporate these findings and discussions into the revised paper. Thank you for your valuable suggestion.
> > >
> > > | RSR-1  | RSR-K  |  ARD   |  PSR   |
> > > | :----: | :----: | :----: | :----: |
> > > | 68.21% | 81.09% | 0.8773 | 81.09% |
> > >
> > > \* This experiment is conducted on class query targeted attack, Siglip-so400m retriever, LLaVA-v1.6-Mistral 7B LVLM, and Places-365 dataset.
> > >
> > >
> > >
> > > A2. Yes, as you suggested, the main contribution of our paper is the class query attack. The single query attack primarily serves as an additional baseline, illustrating the evolution from textual to vision-language RAG attacks and from naive to more realistic assumptions. This helps readers better understand the developmental/derivation trajectory of our method.
> > >
> > >
> > >
> > >
> > > A3. (1) This is a class-type attack. (2) Thank you for your follow-up question, which led us to investigate this phenomenon more deeply. We find that previous examples altered the semantic meaning of the query text (changing the original question's intent), which is the primary reason for the low PSRs. In contrast, when the user query preserves the semantic similarity to a greater extent (e.g., through paraphrasing the original question), the attack effectiveness remains much higher. To verify this, we conduct additional experiments using three target-paraphrased user queries, as shown in the table below. The results demonstrate PSRs around 60%-70%, demonstrating our attack's effectiveness on semantically similar texts.
> > >
> > > |                        User Query                        | Distance to target text | RSR-1  | RSR-K  |  ARD   |  PSR   |
> > > | :------------------------------------------------------: | :---------------------: | :----: | :----: | :----: | :----: |
> > > |    What is the scene category assigned to this image?    |         0.6803          | 59.17% | 72.87% | 0.8019 | 70.13% |
> > > |  Under which scene classification does this image fall?  |         0.7463          | 53.15% | 66.02% | 0.8017 | 57.80% |
> > > | To which scene classification does this picture pertain? |         0.7464          | 53.69% | 66.84% | 0.8014 | 58.08% |
> > >
> > > \* This experiment is conducted on class query targeted attack, Siglip-so400m retriever, LLaVA-v1.6-Mistral 7B LVLM, and Places-365 dataset.
> > >
> > > For example, consider the target text "Which scene category does this image belong to?" compared with the previously used high-relevance text "Which scene does this image represent?". The former focuses on the scene's category/classification, while the latter emphasizes its representative/semantic meaning. These questions address distinct attributes of the scene and therefore modify the original semantic intent to some degree. We will incorporate the above findings and additional experiment results into the revised paper.
> > >
> > >
> > >
> > > A4.  (1) It is conducted on the class type attack. (2) We conduct additional experiments on the poison image with the additional processing to measure its distance to the class center. As shown in the table below, the processed poison image has a distance of 0.7055 to the class center, compared to the average class distance of 0.6521 observed in clean images of the class. Notably, nearly 30% of the class's clean images have distances larger than 0.7055 from the class center, indicating that the poisoned image is still within the class boundary and can be considered to possess representative features of the class.
> > >
> > > | Distance of poison image | Average distance of clean images of the class |
> > > | :----------------------: | :-------------------------------------------: |
> > > |          0.7055          |                    0.6521                     |
> > >
> > > \* This experiment is conducted on enhanced class query targeted attack, Siglip-so400m retriever and Places-365 dataset.

---

### Official Review · Reviewer_41x2 · 2025-03-14

**Overall Recommendation:** 3

**Summary:**

This paper proposes a poisoning attack on Retrieval-Augmented Generation (RAG)-based large vision-language models, enabling the manipulation of outputs for targeted inputs. This is the first study to perform a poisoning attack on a multimodal RAG system. The effectiveness of the two proposed attacks—the single query target attack and the class query targeted attack—has been validated on the OVEN-Wiki database, using Siglip-so400m and CLIP ViT-H as the retrievers.

**Claims And Evidence:**

This work is the first study to explore attacks on RAG-based vision-language model systems. The main idea is to craft a prompt query injected into the RAG database that generates a target response for a given input. Without access to the VLM, the poisoning prompt can be crafted by minimizing the distance between the target query and the poison sample. Extensive experimental results across different VLMs, such as LLaVA and Qwen, show that PoisonedEye achieves a relatively high attack success rate on several classification datasets, including ImageNet-1k, Places-365, and Country-211.

**Essential References Not Discussed:**

I believe the relevant papers have been cited.

**Ethical Review Flag:**

Flag this paper for an ethics review.

**Experimental Designs Or Analyses:**

The experimental settings are reasonable and comprehensive.

**Methods And Evaluation Criteria:**

This work adapts PoisonRAG [1] from LLMs to VLMs. While the attack concept is similar, VLMs require retrieving image-text pairs from the database, which makes crafting the injected query different. This work PoisonedEye needs to minimize both difference between targeting text embedding and poisoning text embedding as well as targeting  image embedding and poisoning image embedding. There are three metrics to quantify the performance of attack in retrieval success: 1) Top-1 retrieval success rate  (RSR); 2) Top-k RSR and 3)  average retrieval distance (ARD).  Poison success rate (RSR) denotes the proportion of target answer occurrences in the responses.


Reference:

[1] Zou, Wei, et al. "Poisonedrag: Knowledge corruption attacks to retrieval-augmented generation of large language models." arXiv preprint arXiv:2402.07867 (2024).

**Other Comments Or Suggestions:**

How much computation is needed to craft the poisoning prompt? Can you provide more details? What is the performance of the attack when there is a limited quota for iterating the poisoning prompt?

**Other Strengths And Weaknesses:**

Please see the above sections.

**Questions For Authors:**

Are you using "I don't know" as the only target response? Do you have experiments using an incorrectly predicted class as the target response?

**Relation To Broader Scientific Literature:**

This is an interesting attempt to extend PoisonRAG to multimodal RAG. However, there is no novel idea in attacking RAG or crafting the poisoning prompt. The contribution is incremental.

**Theoretical Claims:**

There is no theoretical contribution in this work.

---

> ### Author Rebuttal · Authors · 2025-04-01
>
> We sincerely thank your constructive comments.
>
> > This work adapts PoisonRAG [1] from LLMs to VLMs. While the attack concept is similar, VLMs require retrieving image-text pairs from the database, which makes crafting the injected query different.
>
> Thank you for recognizing our contributions and efforts in devising the poisoning attacks on VLMs, which require nontrivial technical development, as you pointed out.
>
> > Q1. How much computation is needed to craft the poisoning prompt? What is the performance of the attack when there is a limited quota for iterating the poisoning prompt?
>
> Since the poison text is fixed, the majority computation in our attack lies in creating the poison image through the signed gradient descent algorithm. A key hyper-parameter to balance computation and poisoning effect is generation steps `s`. The crafted image could converge well when `s` is large enough; however, when `s` is small, the image may not converge well and becomes unstable. We conduct additional ablation experiments to evaluate balance between attack performance and time consumption across different `s`. As shown in the table below, the time required per poison sample is consistently under 20 seconds, demonstrating the efficiency of our attack. Besides, even with only 10 steps, the PSR holds at 69.04%, indicating the effectiveness of our attack under limited iterations.
>
> | Steps | RSR-1  | RSR-K  |  ARD   |  PSR   |  Time  |
> | :---: | :----: | :----: | :----: | :----: | :----: |
> |  10   | 56.43% | 69.31% | 0.8207 | 69.04% | 3.22s  |
> |  20   | 67.12% | 79.17% | 0.8033 | 78.90% | 4.57s  |
> |  50   | 78.08% | 89.31% | 0.7769 | 89.04% | 8.62s  |
> |  100  | 84.10% | 92.60% | 0.7609 | 92.05% | 15.36s |
>
> > Q2. Are you using "I don't know" as the only target response? Do you have experiments using an incorrectly predicted class as the target response?
>
> We have explored 5 different types of target responses and conducted experiments in Section 5.3.3 of our paper. The results show that our attack remains effective on these different target responses. Please refer to Section 5.3.3 and Appendix.G for details. We will further highlight the flexible choice of target response and the corresponding experiments in the revised paper, as you suggested. Thank you for your thoughtful question and valuable advice!

---

> > ### Comment · Reviewer_41x2 · 2025-04-07
> >
> > Thank you for the detailed rebuttal. While some of my concerns have been addressed, I will maintain my original score

---

### Decision · Program_Chairs · 2025-05-01

**Decision:**

Accept (poster)

**Comment:**

This paper adapts the PoisonedRAG from LLM to LVLM with a novel design of method and extensive experiments. All four reviewers have given positive scores with meaningful questions. The authors have addressed these questions through clear explanations and additional experiments. All reviewers like this paper in multiple aspects. Remaining issues: (1) If the AC understands it correctly, the image is perturbed (clean-label) and the text is replaced (noisy-label). However, in Figure 1, it is not clear that the image is perturbed and except in the pseudo-code, it has not been mentioned anywhere how the text and image are manipulated. (2) The AC agrees that MuRAG is somewhat misleading since only vision-language models are considered. The authors should change this term in the final version.